# A Finite Element Investigation on Material and Design Parameters of Ventricular Septal Defect Occluder Devices

**DOI:** 10.3390/jfb13040182

**Published:** 2022-10-09

**Authors:** Zhuo Zhang, Yan Xiong, Jinpeng Hu, Xuying Guo, Xianchun Xu, Juan Chen, Yunbing Wang, Yu Chen

**Affiliations:** 1School of Mechanical Engineering, Sichuan University, Chengdu 610000, China; 2Shanghai Shape Memory Alloy Co., Ltd., Shanghai 200000, China; 3College of Biomedical Engineering, Sichuan University, Chengdu 610000, China; 4Department of Applied Mechanics, Sichuan University, Chengdu 610000, China

**Keywords:** nitinol, polydioxanone, self-expanding occluder, braided wire occluder, finite element analysis (FEA), mechanical performance

## Abstract

Background and Objective: Ventricular septal defects (VSDs) are the most common form of congenital heart defects. The incidence of VSD accounts for 40% of all congenital heart defects (CHDs). With the development of interventional therapy technology, transcatheter VSD closure was introduced as an alternative to open heart surgery. Clinical trials of VSD occluders have yielded promising results, and with the development of new material technologies, biodegradable materials have been introduced into the application of occluders. At present, the research on the mechanical properties of occluders is focused on experimental and clinical trials, and numerical simulation is still a considerable challenge due to the braided nature of the VSD occluder. Finite element analysis (FEA) has proven to be a valid and efficient method to virtually investigate and optimize the mechanical behavior of minimally invasive devices. The objective of this study is to explore the axial resistive performance through experimental and computational testing, and to present the systematic evaluation of the effect of various material and braid parameters by FEA. Methods: In this study, an experimental test was used to investigate the axial resistive force (ARF) of VSD Nitinol occluders under axial displacement loading (ADL), then the corresponding numerical simulation was developed and compared with the experimental results to verify the effectiveness. Based on the above validation, numerical simulations of VSD occluders with different materials (polydioxanone (PDO) and Nitinol with different austenite moduli) and braid parameters (wire density, wire diameter, and angle between left and right discs) provided a clear presentation of mechanical behaviors that included the maximal axial resistive force (MARF), maximal axial displacement (MAD) and initial axial stiffness (IAS), the stress distribution and the maximum principal strain distribution of the device under ADL. Results: The results showed that: (1) In the experimental testing, the axial resistive force (ARF) of the tested occluder, caused by axial displacement loading (ADL), was recorded and it increased linearly from 0 to 4.91 N before reducing. Subsequent computational testing showed that a similar performance in the ARF was experienced, albeit that the peak value of ARF was smaller. (2) The investigated design parameters of wire density, wire diameter and the angle between the left and right discs demonstrated an effective improvement (7.59%, 9.48%, 1.28%, respectively, for MARF, and 1.28%, 1.80%, 3.07%, respectively, for IAS) for the mechanical performance for Nitinol occluders. (3) The most influencing factor was the material; the performance rose by 30% as the Nitinol austenite modulus (EA) increased by 10,000 MPa. The performance of Nitinol was better than that of PDO for certain wire diameters, and the performance improved more obviously (1.80% for Nitinol and 0.64% for PDO in IAS, 9.48% for Nitinol and 2.00% for PDO in MARF) with the increase in wire diameter. (4) For all of the models, the maximum stresses under ADL were distributed at the edge of the disc on the loaded side of the occluders. Conclusions: The experimental testing presented in the study showed that the mechanical performance of the Nitinol occluder and the MARF prove that it has sufficient ability to resist falling out from its intended placement. This study also represents the first experimentally validated computational model of braided occluders, and provides a perception of the influence of geometrical and material parameters in these systems. The results could further provide meaningful suggestions for the design of biodegradable VSD closure devices and to realize a series of applications for biodegradable materials in VSD.

## 1. Introduction

Congenital heart defects (CHD) refer to the general structural abnormality of the heart or thoracic great vessels at birth, with an incidence of 4–50 per 1000 live births [1]. Ventricular septal defect (VSD) is the most common CHD and accounts for about 40% of all CHDs [2]. It occurs in isolation or in combination with other structural defects. Nowadays, with the development of interventional therapy for CHD, the implantation of occlusion devices has become a widely accepted and highly-effective treatment for occluding abnormal blood/thrombus flow within the heart. It is less invasive, avoids extracorporeal circulatory support and surgical scars, and offers a faster recovery [3]. 

Due to their great applicability, a large variety of closure device designs exist on the market employing different types of materials, different geometries, and deployment mechanisms. Currently, the most commonly used clinical interventions are nickel–titanium alloy occluders. Although they have shown good near- to mid-term efficacy, the occurrence of complications is still not considered to be negligible, and there are potential disadvantages and safety risks for metal blockers to remain in the body permanently. The ideal occluder should be biodegradable, and the degradation products should be non-toxic, harmless, and fully absorbed. Polydioxanone (PDO) is a biodegradable polymer which can be completely absorbed in about 6 months, with the degradation products mainly being excreted through urine, and the rest discharged by digestion or as carbon dioxide [4]. PDO has demonstrated in potential applications such as intragastric stents and drug delivery systems [5,6].

The existing work usually evaluates the effectiveness of the mechanical properties of the closure devices through experiments and clinical experience, resulting in a long research and development cycle, high cost, and lack of predictability. Based on experimental rest, Thepphithak et al. [7] evaluated the transformation and mechanical behavior of the nitinol atrial septal defect (ASD) occluder through differential scanning calorimetry (DSC) measurements and pull tests; Wu et al. [8] performed the axial compress in vitro tests of the fully absorbable occluders. And through animal experiments, Huang et al. [9] performed percutaneous transcatheter closure of interventional created VSDs in 16 dogs with the occluders, and obtained gross pathology and histopathology at 6, 12, and 24 weeks of follow-up. Some scholars have validated the application of VSD through clinical experience, such as Tzikas et al. [10], who reported on 19 patients who underwent transcatheter perimembranous ventricular septal defect (PMVSD) closure using a second-generation occluder device, and suggested that the procedure was feasible, safe, and effective. Lee et al. [11] put forward that transcatheter closure of PMVSD with the Amplatzer ductal occlude (ADO) was a safe and promising treatment option, but long-term follow-up in a large number of patients would be warranted.

Numerical simulation represents a powerful support tool providing engineers and clinicians with more detailed information about the effects on the overall biomechanical properties of the material and design parameters of VSD closure devices. Nevertheless, few works focusing on numerical modeling of VSD occlusion devices have been presented; only Li et al. [12] has so far carried out radial compression and axial bending finite element analysis of ventricular septal defect occluders. Nonetheless, only the waist of the device is modeled in a braided structure, while the two discs were simplified into an entity. This work is intended to establish a complete braided numerical model of VSD occluders, and based on the experimental verification of the manufacturer’s product, to provide a reference for subsequent numerical research. 

The stability test is required for occluders to ensure that they remain in place without defect, in accordance with the pharmaceutical industry standards. Also, due to the insufficient mechanical properties of the degradable material, there is a possible problem of insufficient clamping force when the material is changed to a biodegradable material. In this study, the VSD closure device developed by Shanghai Shape Memory Alloy Company was taken as a prototype, which consists of interlaced wires that slide and rotate during deformation. The stability tests were performed as required to investigate the mechanical properties of a commercially available occluder under axial displacement loading (ADL). An accurate finite element model was developed and used to perform computational tests to systematically compare and evaluate the effects on stability of the wire material properties and braid parameters, including the degradability of materials, wire density, wire diameter, and the angle between the left and right discs.

This computational modeling would help to determine the relevant material and design parameters of wire-braided VSD occlusion devices by simulating a wide range of occlusion device configurations. It would further provide references to optimize the VSD occlusion device design. This method could also be further extended to numerical studies of biodegradable occlusion devices and atrial septal occluders. 

## 2. Materials and Methods

### 2.1. Geometry of Commercial VSD Occluder and Modeling

A commercially available occluder (Shanghai Shape Memory Alloy Company, Shanghai, China) for the closure of ventricular septal defects (VSD) was selected as the basic design as shown in Figure 1a, which consists of interlaced wires that slide and rotate as they deform. It is a self-expanding and self-centering occlusion device, prepared by weaving and thermoforming technology using 72 nitinol monofilaments with a diameter of 0.11 mm [13]. The most important part of the occluder device is the double-disc structure, the left and right circular discs, which are linked together by a short connecting waist, as shown in Figure 1b [14]. 

The 3D geometric model was established and imported into ABAQUS2021 software. The complete numerical 3D geometry of the braided occluder was established as shown in Figure 1c. General contact was applied between the wires, with a friction coefficient of 0.25, so that wire-to-wire contact and cross-slip are implemented [15]. 

### 2.2. Experimental Testing

A ventricular septal defect is a birth defect of the heart in which there is a hole in the wall (septum) that separates the two lower chambers (ventricles) of the heart. The occluders are used to plug a hole that is not meant to be present. The device is delivered percutaneously to the hole (the defected site of the heart) in the sheath, and it is then allowed to self-expand to its original double-disc shape when the sheath is removed [16]. After the device is placed properly, the discs of the occluder clamp the VSD, while the waist of the occluder supports the VSD hole. Ideally, the structure of the device will ensure its effective self-placement without falling out, which ensures the stability of the device. However, it is necessary to convincingly prove the effectiveness of the device clamping feature, since fall-out of the occluder would be life-threatening for the patient. Therefore, we designed an experimental test: the occluder was put into the silica model (Figure 2a,b), which was used to simulate the defect, and was then fixed to the test machine, as shown in Figure 2c. After finishing the assembly setup, the loading rod was displaced to make sure it pushed down slowly until the occluder fell out from the silica model (Figure 2d). The value of the axial displacement (AD) and axial resistive force (ARF) of the control rod was recorded.

### 2.3. Computational Testing 

FEA is a lower-cost method to not only replicate experiments but to also carry out repeated tests, finding the most vulnerable point in the model, and optimizing it [17]. A computational framework based on the experimental testing was developed to predict the mechanical performance of wire braid occluders and to explore the role of geometric and material properties on device mechanics. The computational framework was compared and validated by experimental axial testing. A systematic evaluation of the effect of wire braid parameters (e.g., wire density, wire diameter and the angle between left and right discs) and wire materials (Nitinol material properties and biodegradable PDO material) on axial loading response of these devices is presented.

#### 2.3.1. Material Properties

The device is constructed from self-expanding Nitinol mesh, due to its remarkable super-elasticity and shape memory, as well its good biocompatibility and corrosion resistance [18]. The Nitinol occluder is easily crimped at low temperatures and reverts back to its original profile after being released from the sheath at the body temperature [19]. In this study, the Nitinol material was modeled in ABAQUS with a user material subroutine interface (UMAT) [20]. The material properties of this model are shown in Table 1.

The PDO used in the study had a linear elastic behavior in tension, with a Young’s modulus of 1200 MPa and a tensile strength of 326 MPa in the longitudinal direction. For the simulation, the Poisson ratio was set to 0.35.

#### 2.3.2. Boundary and Loading Conditions

All of the FEA processes were undertaken in ABAQUS. The assembly is shown in Figure 3. Besides the occluder, the geometry of the defect and the loading rod were also modeled, which were simplified as rigid bodies controlled by reference points (RP). The translations and rotations of the defective part were limited in all directions. The same limitation was imposed on the loading rod part except for in the Y-axis direction, which applied a displacement as shown in Figure 3a. All the parts had meshed as presented in Figure 3b. The braided occluders were meshed with the 2-nodes linear Timoshenko beam elements (B31), and the defect, as well as loading rod, were both meshed with the 4-nodes 3-D bilinear rigid quadrilateral elements (R3D4). There were 21,886 elements in the occluders, 1513 elements in the defect and 175 elements in the loading rod. Because of the high nonlinearity, a quasi-static analysis was performed by a dynamic explicit algorithm. The quasi-static is guaranteed by making sure that the kinetic energy was less than 5% of the internal strain energy [21].

## 3. Model Validation

As shown in Figure 4a, both the computational testing and experimental testing predicted similar axial resistive force (ARF) trends for braided occluders under axial displacement loading (ADL). Both test results showed ARF increased firstly (from 0 to 4.91 N in experimental testing and from 0 to 4.21 N in computational testing) and decreased later when applying displacement loading. Although the maximum value (14.02% lower) and inflection point (15.6% lower) of the computational testing data are underpredicted, which is a common feature in such models [22], the trend of ARF showed a good agreement with experimental testing, especially in the increasing and stable stage (AD is from 1 to 6 mm of computational testing), and these two stages were the stages we cared most about in this study.

For the comparison of stability characteristics of the device, the initial axial stiffness (IAS) of the occluders at t1 in the increasing stage, as well as the maximal axial resistive force (MARF) and the maximal axial displacement (MAD) at t2 in the stable stage is as shown in Figure 4a. The value of IAS, MARF, and MAD in experimental testing were 0.66 N/mm, 4.91 N, and 7.03 mm, respectively, whereas in the computational testing, they were 0.74 N/mm (overpredicted 12%), 4.21 N (underpredicted 14%) and 5.93 mm (underpredicted 15%), respectively. This error can be partially explained by the complicated deformations of the braided device, where there are substantial self-contacts. Nonetheless, the computational testing method is acceptable to predict the axial response for devices with different parameters.

Figure 4b showed that in moment t1, the occluder clamped the margin of VSD to resist axial loading by deformation until it came to the threshold at moment t2, where the device could barely clamp thereafter. The deformation process is presented in Figure 4c. As the axial loading was applied, the loading disc of the device underwent visible bending and stretching, resulting in transforming the device into concave shape. Moreover, the waist part was elongated and accompanied by a change in the pitch angles.

## 4. Parameter Study

### 4.1. Occluder System Parameter

A systematic evaluation of the effect of geometric and material properties on the functional performance of occluders under the axial loading conditions was carried out by the computational testing method. The commercial wire braid occluder was chosen to be the baseline device geometry with 72 wires of 0.11 mm diameter and an angle of 0° between the left and right disc (two discs parallel to each other). The influence of the Nitinol austenite modulus, wire density, wire diameter, and the angle between the left and right discs were evaluated in wire braid occluders, according to the values shown in Table 2. The model of the occluders with various wire densities (n) and various angles between the left and right discs (α) are presented in Figure 5.

### 4.2. The Results

The effect of key parameters on the axial performance of wire braid occluders is shown in Figure 6.

The results showed that increasing the austenite elastic modulus of Nitinol causes an obvious increase in IAS, MARF, and MAD of the device (increased by 30%, 30%, and 20%, respectively, with every 10,000 MPa increasement), as shown in Figure 6a. It can be seen in Figure 6b that the effects of wire density on the performance cannot be disregarded, as for MARF with a 7.59% linear increase, for IAS with a nonlinear increase of about 1.28% more or less.

The most significant parameter was wire diameter. Figure 6c shows the effects of different wire diameters with PDO and Nitinol materials on the MARF, MAD, and IAS. With the increase in the diameter of the braided wires, the mechanical properties of the occluders made up of both Nitinol and PDO materials had similar trends. The IAS and MARF increased linearly by 1.80% (Nitinol), 0.64% (PDO) and 9.48% (Nitinol), 2.00% (PDO), respectively. The increasing trend in mechanical properties for the PDO material was obviously weaker than that for Nitinol. The performance of the PDO occluders was over 60% lower in IAS and 70% lower in MARF than that of the Nitinol occluders. For a given wire diameter occluder, the mechanical properties of Nitinol were also weaker compared to PDO. The wire diameter of the occluder was 0.07 mm, the IAS of the Nitinol occluder was 0.019 N/mm and that of the PDO was 0.013 N/mm (reduced by 31%). When the wire diameter was 0.09 and 0.11 mm, the difference in IAS was more obvious, which decreased by more than 60%. The MARF of the PDO was also significantly lower than that of Nitinol for these three different wire diameters. In addition, the MAD of the occluder under MARF loading was also different for various braid materials and braiding wire diameters. With the increase in wire diameter, the MAD of the Nitinol occluder decreased slowly, while for the PDO, it was just the reverse, which still needs to be verified and analyzed in subsequent tests.

As shown in Figure 6d, there was also a difference when the angle between the left and right discs appropriately increased, such that, if ignoring the result of the angle at 10°, the IAS and MARF increased by 3.07% and 1.28%, respectively, while the MAD decreased by 2.02% (increased angle resulted in a decrease in disc diameter). The effect of the wire diameter on the performance of braided products was also investigated in terms of both Nitinol and PDO materials.

The deformations of all the devices with MARF after axial loading are shown in Figure 7a–d. The models are divided into 4 groups according to various design parameters from sp1 to sp16 as follows: (a) sp1 to sp4: wire density, (b) sp5 to sp8: Nitinol austenite elastic modulus, (c) sp9 to sp12: wire diameter, and (d) sp13 to sp16: angle between left and right discs of the occluder. In each group, the parameters of the corresponding design variables were changed to explore the different effects of the parameters on the mechanical response under axial displacement load.

The results showed that, except for sp1, all of the models had similar deformations. The margin of the discs, where the maximal stress fields of the device are located, were deformed to resist the loading. All values for maximal stress were over σ^E^_L_ (437.8 MPa, end of the transformation loading for the simulated Nitinol material from austenite to martensite) except sp1, as shown in Figure 7e, which means that the transformations were completely finished in these high-stress fields.

Figure 8 shows the stress-colored map of the maximum axial resistance of the PDO occluder, in which the stress distribution at the edge of the disc on the loaded side of the occluder was significantly larger than that on the rest of the parts. The maximum stress value distributed at the edge of the disc for different braided wire diameters of 0.07, 0.09, and 0.11 mm under the maximum axial load was 21, 26, and 47MPa, respectively. The stress distribution for the NiTi alloy occluder under maximum axial load is shown by the stress-colored map, and presents a similar pattern to that of the PDO occluder, where all of the stress distribution is at the edge of the disc on the loaded side, but the maximum stresses are significantly larger than that of the PDO occluders, which are 542.1 MPa, 568 MPa, 570.3 MPa, respectively.

## 5. Discussion

Nowadays, the implantation of occlusion devices has become widely accepted as a highly effective way to treat abnormal blood/thrombus flow within the heart. Studies of occlusion devices have previously focused on the materials and configurations through experiments and clinical experience, leading to a longer research cycle, high cost and lack of predictability. In this study, not only were the mechanics of the self-expanding wire-braided VSD occluders evaluated through experimental testing, but a systematic evaluation was also developed to investigate the effects of various material and braid parameters using computational simulation.

Based on analytical and experimental testing results, it was demonstrated that axial resistance force would increase to the maximum followed by a decrease, as displacement loading was slowly implemented. The computational method also identified that the stiffness of the device could be effectively improved in at least three aspects, such as the material of the braided wire and the method of weaving, as well as the shape of the hot-pressed double-discs.

Finite element analysis (FEA) has been widely implemented as a productivity tool for design engineers to reduce both development time and cost. It provides a clear presentation of the mechanical response that will help practitioners to avoid mistakes and conduct design improvement for implanted medical devices [23]. However, few studies have applied FEA to the research of occlusion devices. An accurate finite element model would be required to enable the correct estimation of the mechanical behavior of braided closure devices. To the authors’ knowledge, this study represents the first computational model for the braided VSD occluders, which were experimentally validated under ADL and they have great potential in the future design of these devices.

It has been proven that for braided stents, increasing the austenite elastic modulus of Nitinol causes a linear increase in initial radial stiffness [22]. In this study, the FEA testing results showed that for braided occluders, the austenite elastic modulus of Nitinol has the same influence on initial axial stiffness. Nevertheless, the wire density and wire diameter have different effects on initial axial stiffness and initial radical stiffness. For initial radial stiffness, increasing wire density and wire diameter resulted in a linear and non-linear increase, respectively [16,24]. While for initial axial stiffness, this study showed that they cause a non-linear and linear increase separately. The difference might be explained by different deformations under their corresponding loading conditions. During the radial crimping, the straight cylindrical braided stent undergoes a change in the pitch angles, resulting in a diameter reduction as well as a longitudinal elongation of the whole device [25,26]. Applying axial loading, the double-disc braided occluder undergoes obvious bending of the loading disc, besides elongation and change in the pitch angles. The loading disc undergoes concave deformation, the change in the curvature contributes to the increase in bending stress, and eventually a decreased diameter results from being unable to suffer larger loading.

After implanting occlusion devices, the defect will be endothelialized and covered by newly formed autologous tissues. Studies have shown that it takes no longer than 6 months. Therefore, the occluders play a role as a temporary bridge for the self-repair of the heart and should be biodegradable [27]. On the other hand, the pressure difference between the left and right ventricles is about 6.65—14.67 kPa (50—110 mmHg). For the VSD with a diameter of 12 mm, the largest force it would experience from blood would be no more than 1.65 N. It could be noticed that the devices can tolerate a force as large as 9 N and 4.2 N from the experimental and computational testing results, respectively. That means that the Nitinol occluders can effectively seal the defect with a satisfactory mechanical property to avoid falling out. However, it cannot be denied that metal occluders exist in vivo permanently. The weaving material is expected to evolve from non-degradable Nitinol to biodegradable polylatide (PLA), polydioxanone (PDO), polycaprolactone (PCL), etc. These biodegradable materials are non-metallic materials with relatively poor mechanical properties in some aspects, such as initial axial stiffness, compared to Nitinol (lower 30% with same wire diameter). Therefore, new methods for braiding and shaping structures need to be developed to improve the mechanical properties of the biodegradable materials. What has been brought up in this study, through the investigations into the mechanical properties with the FEA method, can be further used for biodegradable materials in the future.

Several limitations and hypotheses of this study need to be mentioned. In computational testing, the defect was assumed to suffer the loading with little deformation, so as to be regarded as a rigid body. The frictional coefficient of all the contact was also assumed to be identical to 0.2. These assumptions were beneficial to FEA numerical computation, but at the risk of losing accuracy. Additionally, the computational testing of parameter studies should also be conducted by in vitro experimental testing to validate the results. When using the biodegradable materials instead of Nitinol to test, the results would be more clinically meaningful. Besides, more testing under different loading conditions should be carried out to fully validate the model and the differences between experimental and FEA testing.

## 6. Conclusions

In summary, this study used experimental tests to investigate the mechanical properties of VSD Nitinol occluders under axial loading and MARF conditions, to prove that they have the sufficient ability to resist falling out from their correct placements. The first experimentally validated computational models of braided occluders were established, and they provided a clear presentation by simulating the process with FEA. Furthermore, the mechanical behavior of different occluder materials and design parameters were also presented by means of FEA. Our results demonstrated that material is the key governing parameter that affects the axial resistive performance of wire-braided occluders. The axial performance of PDO devices is relatively weaker than Nitinol devices; using larger wire diameters and a greater wire density could bring about significant improvement. It is expected that the outcomes of this study could be used as a reference for the future design of biodegradable braided occluders.

## Figures and Tables

**Figure 1 jfb-13-00182-f001:**
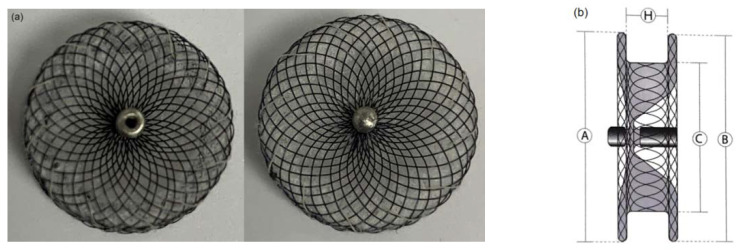
(**a**) VSD occluder product, (**b**) device specification, and (**c**) Numerical 3D model of VSD occluder.

**Figure 2 jfb-13-00182-f002:**
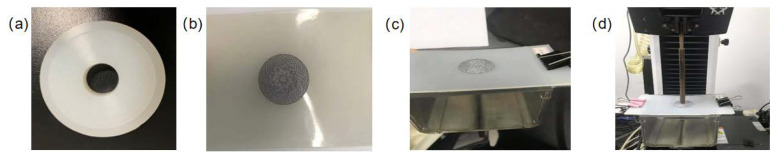
(**a**) Silica defect model, (**b**) silica model blocked by occluder, (**c**) clamp to fix silica model and (**d**) the loading rod to apply axial loading.

**Figure 3 jfb-13-00182-f003:**
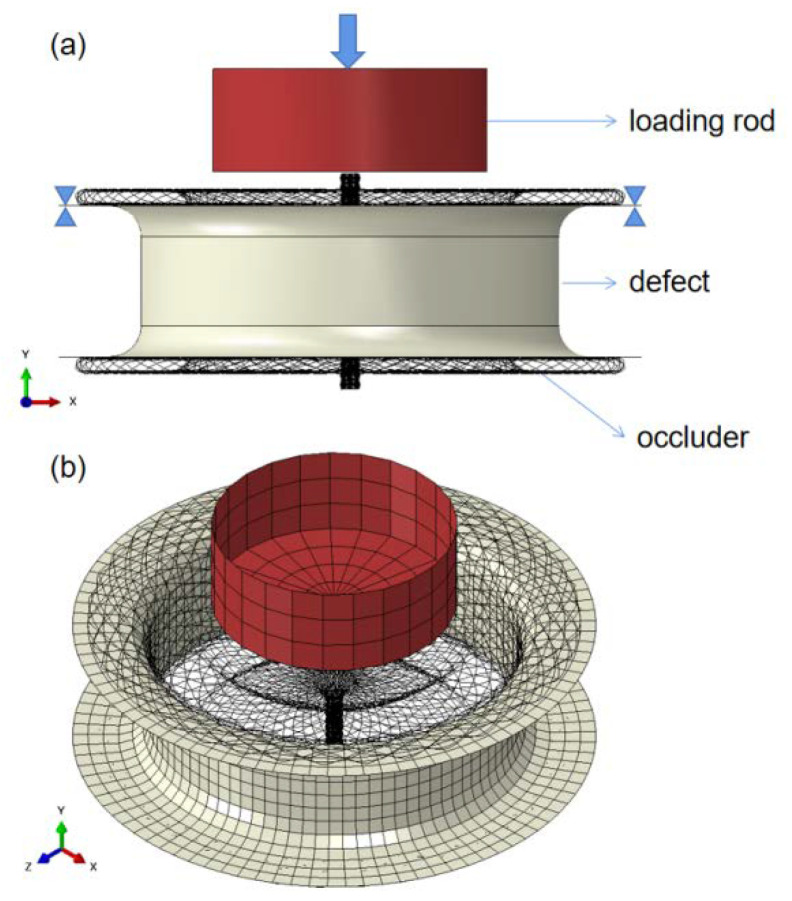
(**a**) Boundary condition and loading, and (**b**) assembly mesh.

**Figure 4 jfb-13-00182-f004:**
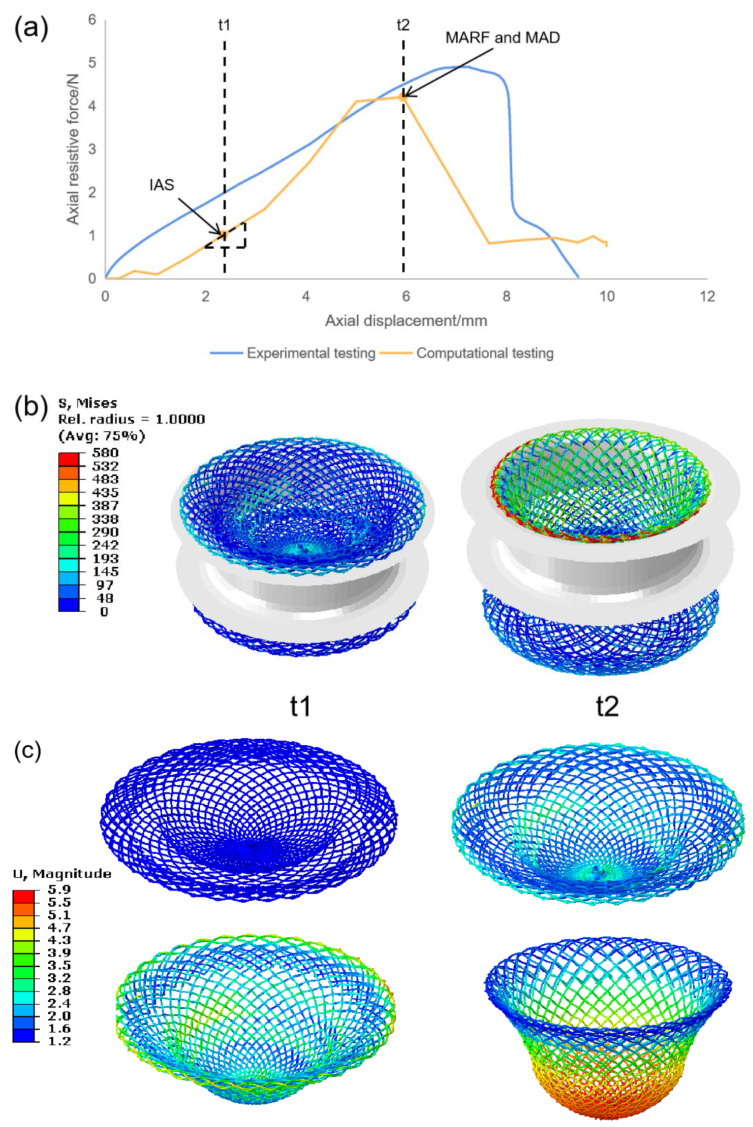
(**a**) Comparison of experimental and computational testing data for devices under ADL. (**b**) Device deformation at t1 and t2. (**c**) The deformation process on the loading discs of the occluder in different displacement loading.

**Figure 5 jfb-13-00182-f005:**
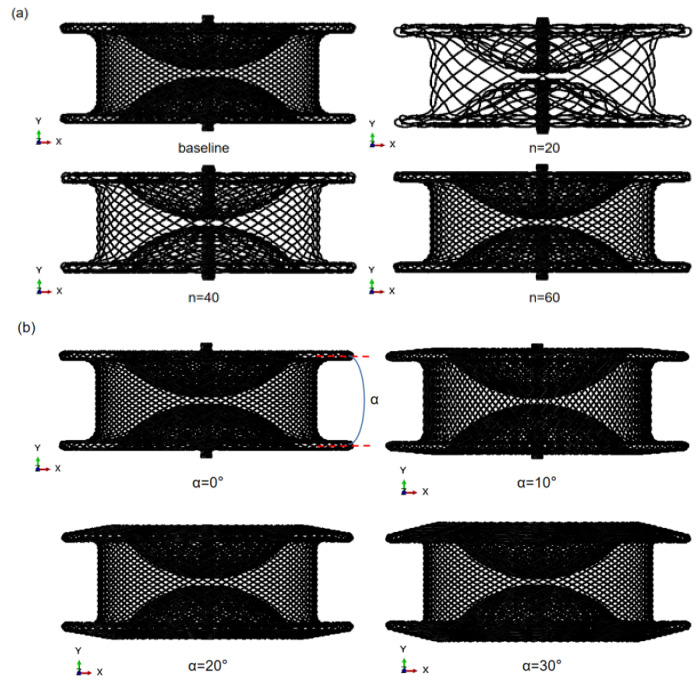
The model with various wire densities (**a**) and various angles between left and right discs (**b**).

**Figure 6 jfb-13-00182-f006:**
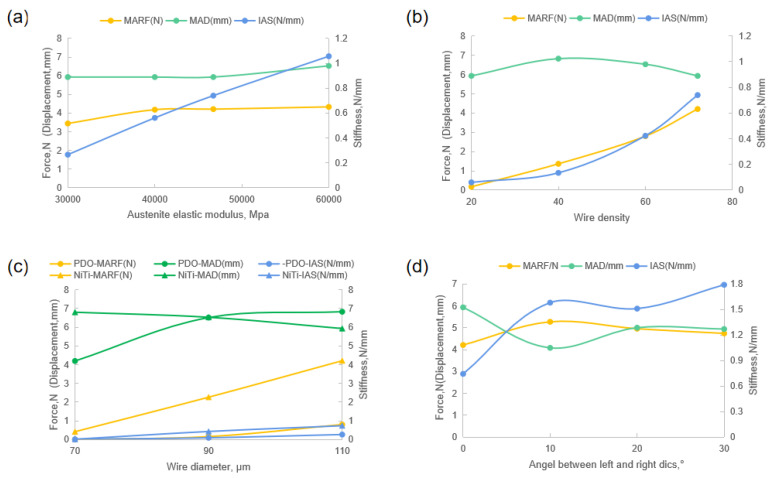
Plots demonstrate the effects of (**a**) Nitinol austenite elastic modulus, (**b**) wire density, (**c**) wire diameter with PDO and Nitinol, and (**d**) angle between left and right discs on MARF, MAD and IAS.

**Figure 7 jfb-13-00182-f007:**
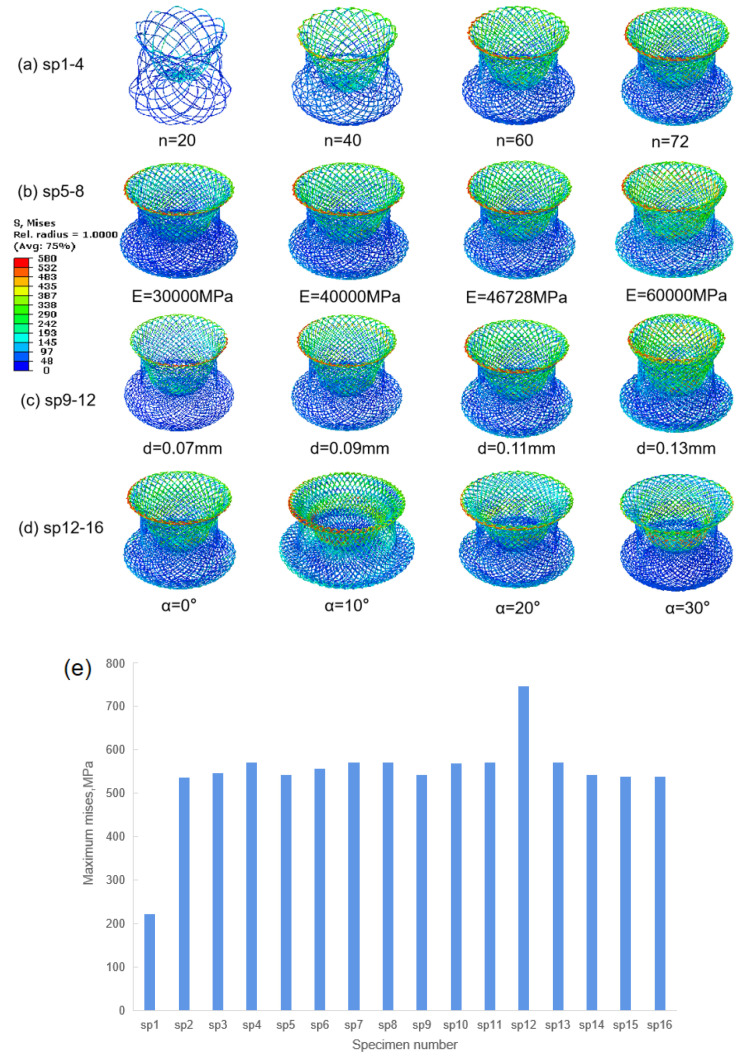
Contour plots showing the effect of, (**a**) wire density, (**b**) Nitinol austenite elastic modulus, (**c**) wire diameter, and (**d**) angle between left and right discs on the stress distribution of the Nitinol device under ADL; (**e**) maximum stress on devices.

**Figure 8 jfb-13-00182-f008:**
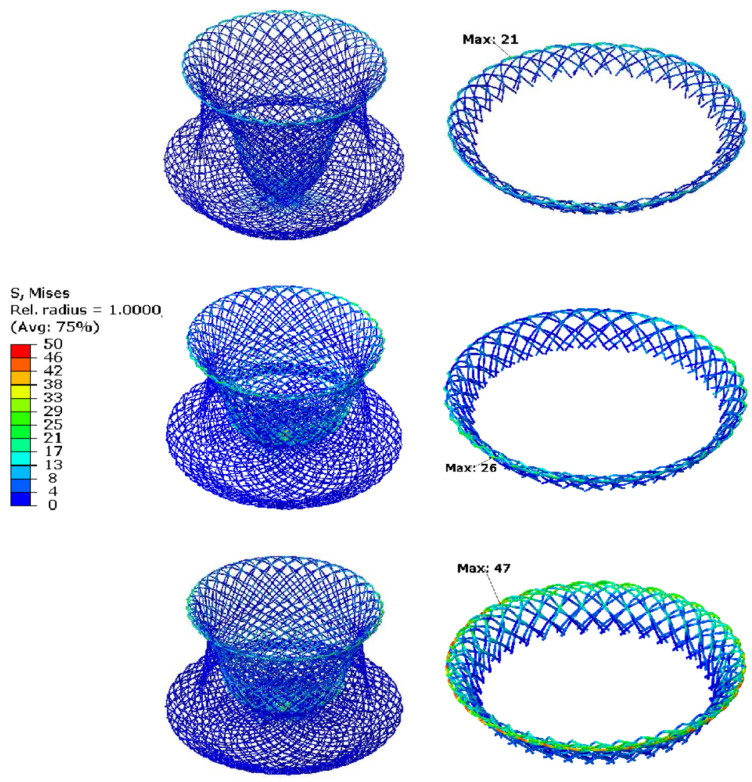
Contour plots showing the effect of wire diameter on the stress distribution of the PDO device under ADL.

**Table 1 jfb-13-00182-t001:** Nitinol material properties.

Symbol	Parameter	Value
E_A_	Austenite elasticity (MPa)	46,728
v_A_	Austenite Poisson’s ratio	0.33
E_M_	Martensite elasticity	25,199
v_M_	Martemosite Poisson’s radio	0.33
ε^L^	Transformation strain	0.0426
(δσ/δT)_L_	(δσ/δT) loading	4.5
σ^S^_L_	Start of transformation loading	358.2
σ^E^_L_	End of transformation loading	437.8
T_0_	Reference temperature	0
(δσ/δT)_U_	(δσ/δT) unloading	4.5
σ^S^_U_	Start of transformation unloading	124.5
σ^E^_U_	End of transformation unloading	17.75
σ^S^_CL_	Start of transformation stress during loading in compression, as a positive value	537.3
ε^L^_V_	Volumetric transformation strain	0.0426

**Table 2 jfb-13-00182-t002:** The baseline parameter properties for a braided occluder and parameter variations.

Wire Parameter	Baseline	Variations	
Nitinol austenite modulus (E_A_)	46,728	30,000; 40,000; 60,000	MPa
Wire density (n)	72	20; 40; 60	wires
Wire diameter (d)	0.11	0.07; 0.09; 0.13	mm
Angle between left and right discs (α)	0	10; 20; 30	°

## Data Availability

Not applicable.

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
