# Peer review of "A Finite Element Investigation on Material and Design Parameters of Ventricular Septal Defect Occluder Devices"

_jfb, 2022, doi:10.3390/jfb13040182_

Round 1
Reviewer 1 Report
The authors present (for me as a cardiologist) very interesting insight into the biomaterials used for VSD occluders. The research question is clear and stringently answered within the manuscript. There are only some minor typos (first sentence after method validation: shown, not show) but English level is sufficiently good overall.
I only have some minor questions or concerns to be answered:
Could you state, what happens to the septum defect when the occluder is fully biodegreedable? Readers might fear, that after degradation the defect is still or again present.
In the results and or conclusion section as well as in the abstract please state which of the occluders you designs was most suitable for its clinical use and why.
Reviewer 2 Report
This study explores the mechanical behavior of Ventricular septal defects occluders with different material and design parameters by Finite elements combined with experimental testing validation.
The paper is well written, but some revisions are necessary:
Is the defect taken into account? The size and geometry of the defect can affect the results.
Paragraph 2.3 is very concise.
The characteristics of the mesh are not clear.
How has the myocardium been characterized?
The results are not clearly written, often some parameters are not described sufficiently.
There is no validation of the results.
Reviewer 3 Report
1. Abstract: (1) Mpa should be MPa. (2) Conclusions should be related to the motivations of the present study, and be supported by the results data.
2. Introduction: (1)The 3rd paragraph, “For example, …… occluders.” This style of description by listing a serial of literatures without logical organization is not a technical form for a paper. Please re-organize them. (2)Please pay attention to the writing of abbreviations. Their formats should be regular, and their full names should be exhibited at their first appearance. (3)“In this study, the VSD closure device……were presented.” Please show the rationale of doing this study. The story should be more logical. (4) Some of the contents are feasible to appear in “Materials and Methods”.
3. Materials and Methods: (1) “The objective of this study is to simulate the experimental testing and predict the mechanical performance of occluders based on different material and design parameters.” This content is feasible to appear in “Introduction”. (2) “the geometry of the defect and the loading rod are also modeled, which are simplified as rigid bodies”. Will this simplification induce significant errors for the simulation compared with the real scenario and experimental results? What are the boundary conditions between the VSD and occluder?
4. Parameter study: (1) “The models are divided into 4 groups from sp1 to sp16”. What are the 4 groups? How are they separated? (2) “cloud diagram”, I suggest the authors use a more professional term.
5. Discussion: (1) “Finite element analysis (FEA)……cardiovascular stent-graft.” This description is well known knowledge, and can be simplified. (2) No NEW results (Fig. 9) are permitted to appear in this section rather than in the Results section.
6. Conclusion: Please summarize the main NEW findings of the present work but not repeat the contents of previously described.
7. The English writing needs native and professional rendering.
